# ADAPTIVE SECOND-ORDER STOCHASTIC OPTIMIZATION

## ABSTRACT

As a much possible way of improving first-order stochastic optimization ($\mathcal{FSO}$), the role of second-order information in stochastic optimization is receiving an increasing attention especially for solving the model with large-scale datasets in recent years, resulting in various second-order stochastic optimization ($\mathcal{SSO}$) methods, e.g., the stochastic Newton (SN) method, the stochastic quasi-Newton (SQN) method, etc. However, the question of how to set an appropriate update rule of the learning rate for SSO methods is still an extremely intractable task, and surprisingly there is quite less literature to tackle this issue. To bridge the gap between the SSO methods and the learning rate, this work develops a class of adaptive SSO methods from the perspective of adaptive gradient methods. Concretely, a general adaptive gradient (GAG) method with the quasi-hyperbolic momentum (QHM) strategy that encompasses Adam, AdaGrad, RMSProp, etc., as the special case of GAG, is incorporated into SN and SQN, respectively, which leads to two methods: SN-GAG and SQN-GAG. In addition, we establish a unified analysis for different adaptive SSO methods, covering their convergence behavior and computational complexity for different backgrounds, such as the strongly convex (SC) case and the Polyak-Łojasiewicz (PŁ) case, where, particularly, the latter is missing in current studies. Finally, numerical tests on different applications of machine learning demonstrate the superiority and the robustness of the resulting methods.

## 1 INTRODUCTION

Considering the stochastic optimization problem shown below which has the most broadly type in large-scale machine learning (ML), natural language processing (NLP), artificial intelligence (AR), etc.,

$$\min_{x \in \mathbb{R}^d} F(x) = \mathbb{E}[f(x, \xi)], \tag{1}$$

where $f : \mathbb{R}^d \to \mathbb{R}$ is continuously but possibly non-convex, $\xi$ represents a random variable with distribution $\mathcal{P}$, and $\mathbb{E}[\cdot]$ represents the expectation taken with respect to $\xi$. In general, the objective function $f(\cdot, \xi)$ is not given explicitly or the distribution $\mathcal{P}$ is unknown, making it difficult to compute the function value and its gradient. Practically, one often approximates the model (1) by the empirical risk minimization problem (Huang & Pu, 2022; Vlaski & Sayed, 2022; Nemeth & Fearnhead, 2021; Bonnabel, 2013),

$$\min_{x \in \mathbb{R}^d} F(x) = \frac{1}{n} \sum_{i=1}^n f_i(x), \tag{2}$$

where $f_i : \mathbb{R}^d \to \mathbb{R}$ is the loss function that corresponds to the $i$th data sample selected from a collection of independent and identically distributed samples, and $n$ denotes the number of data samples which is supposed to be extremely large.

Gradient descent (GD)-based optimization methods are a popular type of methods to solve model (2), where it often employs the following iterative scheme as shown in (3). Nevertheless, the evaluation of full gradient, $\nabla F(x) = \sum_{i=1}^n \frac{1}{n} \nabla f_i(x)$, is expensive especially for large-scale model, where $n$ is huge. It is necessary to use stochastic optimization methods (also familiar with stochastic approximation methods) to address model (2), which was firstly developed by the pioneering work

of Robbins and Monro (Robbins & Monro, 1951). The leading methodology in different applications of ML, NLP, AR, etc., advocates utilizing stochastic gradient descent (SGD) methods (Bottou, 2012). In the $k$th update step, SGD selects a subset $\mathcal{S} \subset \{1, 2, \cdots, n\}$ with $B$ samples and then evaluates the stochastic gradient estimator $\nabla F_{\mathcal{S}}(x_k)$ as described in (3),

$$x_{k+1} = x_k - \eta_k g_k, g_k = \left\{ \begin{array}{l} \nabla F(x_k) = \frac{1}{n} \sum_{i=1}^{n} \nabla f_i(x_k), \text{(GD)} \\ \nabla F_{\mathcal{S}}(x_k) = \frac{1}{B} \sum_{i \in \mathcal{S}} \nabla f_i(x_k), \text{(SGD)} \end{array} \right. \tag{3}$$

where $\eta_k > 0$ denotes the $k$th learning rate. $\nabla F_{\mathcal{S}}(x_k)$ is usually an unbiased estimator of the gradient of $F(x)$ at $x_k$ that is $\mathbb{E}[\nabla F_{\mathcal{S}}(x_k)] = \nabla F(x_k)$. If we adopt $B = 1$, the iterative scheme (3) falls into vanilla SGD.

SGD usually converges slowly and is greatly sensitive to hyper-parameter settings due to high variance. Many techniques, including but not limited to momentum, second-order information, importance sampling, variance reduction, and adaptive learning rates, have been proposed to solve worse performance of SGD. Among these techniques, SGD with momentum is broadly employed, especially in deep learning. Via automatically acquiring the learning rate for SGD, adaptive learning rates, such as the Barzilai-Borwein technique (Barzilai & Borwein, 1988), the Polyak learning rate (Ren et al., 2022), the hyper-gradient descent technique (Baydin et al., 2018), AdaGrad (Duchi et al., 2011), Adam (Kingma, 2014), RMSProp (Tieleman et al., 2012), AMSGRAD (Reddi et al., 2018), are another continually being discussed and updated technique. From the side of manipulating variance of stochastic optimization methods, stochastic variance reduction methods, involving SAG (Roux et al., 2012), SAGA (Defazio et al., 2014), SVRG (Johnson & Zhang, 2013), SARAH (Nguyen et al., 2017a), SPIDER (Fang et al., 2018), SCSG (Lei et al., 2017), etc., attain a linear convergence rate for the strongly convex (SC) model. In contrast, SGD with second-order information shows its superiority on highly nonlinear and ill-conditional problems by adapting to the curvature of the problem.

Second-order stochastic optimization (SSO) methods solve the impractical of evaluating gradient and Hessian matrix exactly in second-order deterministic optimization methods especially for large-scale optimization. For instance, Xu et al. (2020) proposed the trust region method with inexact Hessian, where the second-order information was approximated via the subsampled Hessian matrix, but the gradient was still evaluated exactly. Kohler & Lucchi (2017) developed a stochastic version of adaptive regularization using cubics (ARC), yet they need a much stronger assumption in both gradient and Hessian approximation. Other well-known second-order stochastic optimization (SSO) methods are the stochastic Newton (SN) method, the stochastic quasi-Newton (SQN) method and their variants. The iterative scheme of stochastic version of Newton-like methods is generally reformulated as:

$$x_{k+1} = x_k - \eta B_k^{-1} g_k, \tag{4}$$

where (4) is obtained by evaluating the minimizer of a second-order Taylor series approximation as follows:

$$F_{\mathcal{S}_H}(x) = F_{\mathcal{S}_H}(x_k) + \nabla F_{\mathcal{S}_H}(x_k)^T(x - x_k) + \frac{1}{2}(x - x_k)^T B(x - x_k), \tag{5}$$

where $\mathcal{S}_H \subseteq [n]$ with $|\mathcal{S}_H| = B_H$. If $B = B_k = \nabla^2 F_{\mathcal{S}_H}(x_k)$, (4) turns to the canonical SN method. In contrast, if $B = B_k$ is some approximation generated basing on stochastic gradient, (4) falls to the SQN method. More specifically, if $B = B_k = I$, (4) goes to vanilla SGD.

## 1.1 RELATED WORK

**Conventional $\mathcal{SSO}$ Methods.** The work in the literature (Schraudolph et al., 2007; Yousefian et al., 2016; Mokhtari & Ribeiro, 2014; Byrd et al., 2016) developed various SQN-type methods, but have not been entirely successful, where the convergence rate of early stages of SQN-type algorithmic framework is only sub-linear. Obviously, the theoretical performance of stochastic version of Newton-like algorithms is not better than that of SGD. The works in (Yousefian et al., 2016; Mokhtari & Ribeiro, 2014) considered the stochastic version of the Broyden-Fletcher-Goldfarb-Shanno (BFGS) framework. In the work (Chen et al., 2019), the authors discussed the stochastic version of the BFGS framework and the limited Broyden-Fletcher-Goldfarb-Shanno (L-BFGS) framework simultaneously. In contrast, the studies in (Schraudolph et al., 2007; Byrd et al., 2016)

adopted L-BFGS framework. Bach & Moulines (2013) developed the two-stage online Newton technique, where the first stage executed average SGD with the learning rate of order $O\left(\frac{1}{\sqrt{k}}\right)$, and the second stage optimized a quadratic model of the loss function with a constant learning rate. By combining stochastic semismooth Newton steps and stochastic proximal gradient steps, Milzarek et al. (2019) developed a globalized stochastic semismooth Newton method for addressing stochastic optimization problems involving smooth non-convex and non-smooth convex functions.

**Variance Reduction for $\mathcal{SSO}$ Methods.** Based on variance-reduced techniques, various faster SQN-type methods have been developed, including the incremental quasi-Newton (IQN) method (Mokhtari et al., 2018), the stochastic limited Broyden-Fletcher-Goldfarb-Shanno (SLBFGS) method (Moritz et al., 2016), VR-MZ-SQN (Chen & Feng, 2023), the linear time stochastic second-order algorithm (LiSSA) (Agarwal et al., 2017), SpiderSQN (Zhang et al., 2021), and the stochastic variance-reduced cubic regularized Newton method (SVRC) (Zhou et al., 2018). Particularly, Moritz et al. (2016) showed the linear convergence rate of SLBFGS on large-scale convex and non-convex optimization problems. In addition, Gower et al. (2016) developed a stochastic block L-BFGS method with variance reduction and demonstrated its linear convergence rate. Zhang et al. (2023) developed a general framework that introduced decentralized SQN with variance reduction to realize fast convergence. Kasai et al. (2019) developed a Riemannian SQN method with variance reduction. Zhu et al. (2020) developed a new variance reduction and quasi-Newton preconditioning framework for particle-based variational inference methods.

**Momentum Techniques for $\mathcal{SSO}$ Methods.** The role of momentum in $\mathcal{SSO}$ methods has been investigated by many studies. To improve the practical performance of SpiderSQN, Zhang et al. (2021) incorporated different momentum schemes into SpiderSQN. Yasuda et al. (2019) put forward the stochastic variance-reduced Nesterov's accelerated quasi-Newton methods in full and limited memory forms. Similarly, Indrapriyadarsini et al. (2020) developed the SQN method with Nesterov's accelerated gradient (NAG) in both its full and limited memory forms for dealing with large-scale non-convex optimization problems in neural networks. Makmuang et al. (2023) came up with the regularized stochastic Nesterovs accelerated quasi-Newton method to effectively accelerate the convergence rate and avoid the near-singularity problem of the Hessian update in the stochastic BFGS method.

## 1.2 MAIN CONTRIBUTIONS

For first-order stochastic optimization ($\mathcal{FSO}$) algorithms, various update rules of the learning rate have been proposed as mentioned above. Surprisingly, the research on the role of the learning rate in $\mathcal{SSO}$ methods is quite limited. The existing $\mathcal{SSO}$ algorithms usually work with a scalar constant learning rate, or a diminishing learning rate (Zhang et al., 2021; Zhu et al., 2020). Also, the line search technique is also considered in $\mathcal{SSO}$ algorithms to obtain the learning rate (Guo et al., 2023; Wills & Schön, 2021; Byrd et al., 2012; Schraudolph et al., 2007). However, most of them are time consuming or impractical for large-scale models. In addition, we found that Duchi et al. (2011) applied AdaGrad to compute the learning rate for SQN. Instead of using the line search technique, Zhou et al. (2017) proposed using the properties of self-concordant functions to compute an adaptive learning rate for BFGS and thereby avoided executing line searches. To bridge the gap between $\mathcal{SSO}$ methods and the learning rate, this work equips second-order stochastic optimization methods with an adaptive update rule of the learning rate. For clarity, we summarize our main contributions as follows:

(1) We develop a class of adaptive second-order stochastic optimization methods by utilizing a general adaptive gradient (GAG) method to compute the learning rate for $\mathcal{SSO}$ methods, where GAG encompasses most of existing adaptive gradient methods, such as AdaGrad, RMSProp, Adam, etc. Specifically, we incorporate such the learning rate into classical SN method and the SQN method, respectively, leading to two novel $\mathcal{SSO}$ methods, referred to as SN-GAG and SQN-GAG.

(2) Further, we establish a unified analysis for SN-GAG and SQN-GAG under different backgrounds, involving the strongly convex (SC) objective function and the Polyak-Łojasiewicz (PŁ) objective function. Particularly, under mild conditions, we prove that the resulting SN-GAG and SQN-GAG methods have a linear convergence rate and recover the well-known oracle complexity for models with the SC and PŁ constraints respectively.

(3) Finally, our empirical analysis on different machine learning tasks demonstrates that the resulting algorithms perform better in contrast to classical adaptive gradient methods, state-of-the-art $\mathcal{FSO}$ methods, and $\mathcal{SSO}$ methods. Moreover, various numerical tests show the robustness of our methods to different key hyper-parameters.

## 2 PRELIMINARIES

### 2.1 BASIC NOTATIONS

Throughout this work, for a vector $x$, $x^T$ denotes its transpose, while $\|x\|$ represents the Euclidean vector norm that is $\|x\| = \sqrt{x^T x}$. We denote $x_* = \arg\min F(x)$. We denote the identity matrix by I. We write $\mathbb{E}[z]$ as the expectation of the random variable $z$ and denote $[n] = \{1, \cdots, n\}$. We write $\nabla F_{\mathcal{S}}(x) = \frac{1}{B} \sum_{i \in \mathcal{S}} \nabla f_i(x)$ and $\nabla F_{\mathcal{S}_H}(x) = \frac{1}{B_H} \sum_{i \in \mathcal{S}_H} \nabla f_i(x)$, where $[\mathcal{S}] = \{1, 2, \ldots, \mathcal{S}\}$ with $B$ samples and $[\mathcal{S}_H] = \{1, 2, \ldots, \mathcal{S}_H\}$ with $B_H$ samples. We write $\nabla F(x)$ and $\nabla^2 F(x)$ as the gradient and Hessian matrix of the objective function $F(x)$, respectively. Considering two sequences, $\{a_n\}$ and $\{b_n\}$, if there exists a constant $C > 0$ such that $a_n \leq Cb_n$, we remark $a_n = O(b_n)$. For two matrices $A$, $B$ of the same dimensions, we use $A \succ 0$ to indicate $A$ is positive definite; $A \prec B$ to indicate that $B - A \succ 0$.

### 2.2 ADAPTIVE GRADIENT METHODS

To better understand this paper, we, here, introduce several popular adaptive gradient methods. The first one, we will introduce, is AdaGrad, where it is proposed by (Duchi et al., 2011) and adopts the following iterative scheme:

$$\textbf{AdaGrad} : \begin{cases} m_k = g_k, \\ u_k = u_{k-1} + g_k^2, \\ x_{k+1} = x_k - \eta_k m_k / (\sqrt{u_k} + \epsilon), \end{cases} \tag{6}$$

where $\epsilon \geq 0$. Note that, for clarity and convenience, we call $\eta_k$ in an adaptive gradient method the base learning rate and $\eta_k / \sqrt{u_k}$ the effective learning rate. While AdaGrad confirmed effectively for sparse optimization problems, experiments showed that AdaGrad-type methods perform worse when the objective function is non-convex and gradients of the objective function are dense.

To deal with the issue in AdaGrad-type methods, RMSProp proposed using an exponential moving average rather than a cumulative sum, where the iterative scheme of RMSProp is formulated as

$$\textbf{RMSProp} : \begin{cases} m_k = g_k, \\ u_k = \beta u_{k-1} + (1 - \beta)g_k^2, \\ x_{k+1} = x_k - \eta_k m_k / (\sqrt{u_k} + \epsilon), \end{cases} \tag{7}$$

where $\beta \in (0, 1)$.

Further, utilizing the idea of RMSProp and adding the heavy-ball like momentum into the first moment estimate, Kingma (2014) proposed Adam, working with the iterative scheme below:

$$\textbf{Adam} : \begin{cases} m_k = \beta_1 m_{k-1} + (1 - \beta_1)g_k, \\ u_k = \beta_2 u_{k-1} + (1 - \beta_2)g_k^2, \\ x_{k+1} = x_k - \eta_k m_k / (\sqrt{u_k} + \epsilon), \end{cases} \tag{8}$$

where $\beta_1 \in (0, 1)$ and $\beta_2 \in (0, 1)$.

For most of existing adaptive gradient methods, only partial convergence results are established. More specifically, since several studies (Rubio, 2017; Reddi et al., 2018) pointed out the issues in the proof of Adam (shown in (Kingma, 2014)), many authors pay much attention to the convergence behavior of adaptive gradient methods. In contrast, this work provides a theoretical guarantee for $\mathcal{SSO}$ methods with different adaptive gradient methods.

### 2.3 ASSUMPTIONS

In this work, we study adaptive $\mathcal{SSO}$ methods for the model with the SC and PŁ constraints, respectively. We now provide some assumptions that are necessary in this work.

**Assumption 1.** *The loss function $F(x)$ and its gradient, in model (2) satisfy the following properties:*

> *(a)* (**Lipschitz Continuous**) *$F(x) : \mathbb{R}^d \to \mathbb{R}$ is continuously differentiable and has L-lipschitz gradient, i.e., for $\forall x, y \in \mathbb{R}^d$*

$$\|\nabla F(y) - \nabla F(x)\| \leq L\|y - x\| \tag{9}$$

> *(b)* (**PŁ Condition**) *The objective function $F(x)$ satisfies PŁ condition if $\exists \delta > 0$, for $\forall x \in \mathbb{R}^d$, such that*

$$F(x) - F(x_*) \leq \delta\|\nabla F(x)\|^2, \tag{10}$$

> *where $x_* = \arg\min F(x)$.*

> *(c)* (**Bound**) *For any iteration $k$, we have*

$$\mathbb{E}\left[\|\nabla f_i(x_k)\|^2\right] \leq \gamma^2, \tag{11}$$

$$\mathbb{E}\left[\|\nabla F_{\mathcal{S}}(x_k) - \nabla F(x_k))\|^2\right] \leq \frac{\sigma^2}{B}, \tag{12}$$

> *where $\sigma > 0$ denotes the noise level of the gradient estimator and $\mathcal{S} \subset [n]$ with $B$ samples.*

**Assumption 2.** *There exist two positive constants $\lambda$ and $\Lambda$ such that for $\forall x \in \mathbb{R}^d$*

$$\lambda I \prec \nabla^2 F_{\mathcal{S}_H}(x) \prec \Lambda I, \tag{13}$$

*where $\mathcal{S}_H \subset [n]$ with $B_H$ samples.*

*The result in (13) implies that the function $F(x)$ in model (2) keeps the following conclusion:*

$$\lambda I \prec \nabla^2 F(x) \prec \Lambda I \tag{14}$$

For the $L$-smooth objective function $F(x)$, it admits the following equivalent form, for all $\forall x, y \in \mathbb{R}^d$,

$$F(y) \leq F(x) + \langle \nabla F(x), y - x \rangle + \frac{L}{2}\|y - x\|^2. \tag{15}$$

We also establish the convergence guarantee of the methods for the SC case, i.e., the objective function satisfies the following assumption:

**Assumption 3.** *(**SC Condition**) A differential objective function $F(x)$ is $\mu$-strongly convex if $\forall x, y \in \mathbb{R}^d$*

$$F(y) \geq F(x) + \langle \nabla F(x), y - x \rangle + \frac{\mu}{2}\|y - x\|^2. \tag{16}$$

According to the strong convexity of the objective function $F(w)$, we have that for $\forall x \in \mathbb{R}^d$

$$2\mu[F(x) - F(x_*)] \leq \|\nabla F(x)\|^2, \tag{17}$$

where $x_* = \arg\min F(x)$.

## 3 STOCHASTIC NEWTON ALGORITHMS WITH GENERAL ADAPTIVE GRADIENT

This section considers the classical SN method with GAG for solving model (2), involving the SC case and the PŁ case. We first introduce our SN-GAG method in subsection 3.1. Subsequently, we provide a theoretical guarantee of our SN-GAG method in subsection 3.2 for different cases.

## 3.1 SN-GAG

The details of GAG are discussed here. GAG introduces the quasi-hyperbolic momentum (QHM) technique (Gitman et al., 2019) into adaptive gradient methods, where the update scheme of QHM is shown in (18).

$$\mathbf{QHM} : \begin{cases} m_t = \beta m_{t-1} + (1 - \beta)g_t, \\ x_t = x_{t-1} - \eta[\xi m_t + (1 - \xi)g_t], \end{cases} \tag{18}$$

where $\xi \in [0, 1]$. We establish the relationship between QHM and the several existing momentum techniques. For example, if $\xi = \beta$, QHM falls into NAG. While if $\xi = 1$, QHM becomes heavy-ball momentum (HBM) (Polyak, 1964). Additionally, if $\xi = 0$, the update scheme (18) turns to plain SGD.

Algorithm 1 describes the details of our first adaptive $\mathcal{SSO}$ method, SN-GAG.

---

**Algorithm 1** SN-GAG

---

**Require:** base learning rate $\eta$, outer loop size $\mathfrak{S}$, inner loop size $\mathfrak{K}$, batch sample $B$, the preconditioned parameters $\epsilon, \eta, \alpha_1, \alpha_2, \beta_1$, and $\beta_2$
    **Initialize:** $\widetilde{x}^0$, $G_0^0 = \mathbf{0}$, and $U_0^0 = \mathbf{0}$
    **for** $s = 1$ **to** $\mathfrak{S}$ **do**
        $\widetilde{x} = x_0^s = \widetilde{x}^{s-1}$
        $g^s = \nabla F(\widetilde{x})$
        **for** $k = 1$ **to** $\mathfrak{K}$ **do**
            Select a sample $\mathcal{S} \subset [n]$, with $|\mathcal{S}| = B$, and Compute first-order stochastic gradient estimator, $V_k^s$, $V_k^s = \nabla F_{\mathcal{S}}(x_{k-1}^s) - \nabla F_{\mathcal{S}}(\widetilde{x}) + g^s$
            Select $\mathcal{S}_H \subset [n]$, with $|\mathcal{S}_H| = B_H$, and update the Newton direction by using Hessian matrix, $\nabla^2 F_{S_H}(x_k^s)$,
            **if** $k < 1$ **then**
                $V_{\mathrm{ND}}^s = V_k^s$
            **else**
                $V_{\mathrm{ND}}^s = (\nabla^2 F_{S_H}(x_k^s))^{-1} V_k^s$
            **end if**
            Compute momentum
        $G_k^s = \beta_1 G_{k-1}^s + (1 - \beta_1) V_{\mathrm{ND}}^s$
            Compute adaptive learning rate
        $U_k^s = \beta_2 U_{k-1}^s + (1 - \beta_2)(\nabla F_{\mathcal{S}}(x_{k-1}^s))^2$
            Update parameters
        $x_k^s = x_{k-1}^s - \eta \cdot \frac{\alpha_1 G_k^s + (1 - \alpha_1) V_{\mathrm{ND}}^s}{\sqrt{\alpha_2 U_k^s + (1 - \alpha_2)(\nabla F_{\mathcal{S}}(x_{k-1}^s))^2 + \epsilon}}$
        **end for**
        $\widetilde{x}^s = x_{\mathfrak{K}}^s$
    **end for**

---

**Remark 1:** For SN-GAG (Algorithm 1), we have the following remarks:

(1) In SN-GAG (Algorithm 1), the SVRG gradient estimator, $V_k^s = \nabla F_{\mathcal{S}}(x_{k-1}^s) - \nabla F_{\mathcal{S}}(\widetilde{x}) + \frac{1}{n} \sum_i^n \nabla f_i(\widetilde{x})$, is employed. For the SVRG gradient estimator, we have that it is an unbiased stochastic gradient estimator due to $\mathbb{E}[V_k^s] = \nabla F(x_{k-1}^s)$. Actually, other types of stochastic gradient estimators, such as SARAH, SAG, SAGA, SPIDER, etc., can also be introduced into SN-GAG (Algorithm 1).

(2) SN-GAG (Algorithm 1) has some certain connections with most of existing adaptive gradient methods. For instance, if $\alpha_1 = 0$ and $\alpha_2 = 1$, we equip the RMSProp-like method into SN. While if $\alpha_1 = \alpha_2 = 1$, we introduce the Adam-like method into SN. Additionally, if $\alpha_1 = 1$ and $\alpha_1 = \beta_1$, the NAdam-like method is incorporated into SN.

(3) In general, the Hessian matrix, $\nabla^s F_{\mathcal{S}_H}(x_k^s)$, is sometimes nearly singular, especially when the sample size is quite small. To avoid this case, the update direction, $V_{\mathrm{ND}}^s = (\nabla^2 F_{S_H}(x_k^s))^{-1} V_k^s$, in SN-GAG (Algorithm 1) is replaced by $\mathcal{V}_{\mathrm{ND}}^s = (\nabla^2 F_{S_H}(x_k^s) + \xi \mathrm{I})^{-1} V_k^s$ in practice, where $\xi$ is a positive constant.

## 3.2 CONVERGENCE ANALYSIS FOR SN-GAG

The theoretical guarantee of SN-GAG (Algorithm 1) for the SC case and the PŁ case is established in this segment. Concretely, the theoretical result of SN-GAG (Algorithm 1) for the SC case is provided in subsection 3.2.1. The main theoretical result of SN-GAG (Algorithm 1) for the PŁ case is offered in subsection 3.2.2, respectively.

### 3.2.1 CONVERGENCE PROPERTY ON THE SC CASE

The theoretical result of SN-GAG (Algorithm 1) for the SC objective function is established in Theorem 1.

**Theorem 1.** *Let Assumption 1(a), Assumption 1(c), and Assumption 2 hold and $x_* = \arg\min_{x \in \mathbb{R}^d} F(x)$. Choose $\mathcal{S} \subseteq [n]$ with $B$ samples and $\mathcal{S}_H \subseteq [n]$ with $B_H$ samples, set $\eta = \frac{1}{2L(1-\alpha_1\beta_1)}$, where $\alpha_1 \in [0,1]$ and $\beta_1 \in (0,1)$ and assume that $\mathfrak{K}$, $\alpha_1$, $\beta_1$ are further picked up so that*

$$\rho = \frac{2L(1-\alpha_1\beta_1)}{\mu\mathfrak{K}} < 1. \tag{19}$$

*Then SN-GAG (Algorithm 1) attains a linear convergence speed in expectation with rate $\rho$:*

$$\mathbb{E}\left[\|\nabla F(\tilde{x}^{\mathfrak{S}})\|^2\right] \leq \rho^{\mathfrak{S}} \|\nabla F(\tilde{x}^0)\|^2. \tag{20}$$

*Proof.* Technical proofs are available in Appendix B.1. □

Note that in order to acquire a linear convergence rate, it is necessary to execute $\mathfrak{S} = O(\log(1/\varepsilon))$ outer loop sizes. Additionally, to keep $\rho < 1$, one needs to ask for $\mathfrak{K} = 2L(1-\alpha_1\beta_1)/\mu$. As a result, we have the overall gradient complexity of SN-GAG (Algorithm 1) for the model with the SC constraint is $(n + (2B + B_H)\mathfrak{K})\mathfrak{S}$, i.e.,

$$O\left((n + 2\kappa(2B + B_H)(1-\alpha_1\beta_1))\log\left(\frac{1}{\varepsilon}\right)\right), \tag{21}$$

where $\kappa = L/\mu$ usually denotes the condition number.

To follow this result easily, we point out the complexity of modern $\mathcal{FSO}$ methods and $\mathcal{SSO}$ methods for dealing with models with the SC constraint. For instance, in order to solve the model (2) with the SC constraint, SVRG, SARAH, SAG, and SAGA (variants of $\mathcal{FSO}$ methods) require

$$O\left((n + \kappa)\log\left(\frac{1}{\varepsilon}\right)\right) \tag{22}$$

gradient computations in finding an $\varepsilon$-accurate solution.

For $\mathcal{SSO}$ methods, utilizing the SVRG-like gradient estimator, Lucchi et al. (2015) proposed the variance-reduced stochastic Newton method (coined VITE) and showed that the gradient complexity of VITE is

$$O\left((n + (B + B_H)\mathfrak{K})\log\left(\frac{1}{\varepsilon}\right)\right). \tag{23}$$

Similarly, utilizing the SVRG-like gradient estimator, Zhang et al. (2023) designed two fully decentralized SQN methods, the damped regularized limited-memory Davidon-Fletcher-Powell (DFP) and the damped limited-memory BFGS, where they showed that the complexity of these two $\mathcal{SSO}$-like methods for solving the model (2) with the SC constraint is

$$O\left(\left(m + \frac{B\kappa^2\kappa_H^2 \log\frac{\kappa\kappa_H}{1-\Lambda^2}}{(1-\Lambda^2)^2}\right)\log\left(\frac{1}{\varepsilon}\right)\right), \tag{24}$$

where $m$ denotes the number of local samples, $\kappa_H = M_1/M_2$ denotes the number of the Hessian inverse approximation, and $1-\Lambda^2$ denotes the connectedness of the network.

Therefore, we safely summarize that apart from the linear convergence rate, the computational complexity of SN-GAG (Algorithm 1) is comparable to that of modern $\mathcal{FSO}$ and $\mathcal{SSO}$ methods.

### 3.2.2 CONVERGENCE PROPERTY ON THE PŁ CASE

Further, the convergence behavior of SN-GAG (Algorithm 1) for the model (2) with the PŁ objective function is provided in Theorem 2.

**Theorem 2.** *Let Assumption 1 hold and $x_* = \arg\min_{x \in \mathbb{R}^d} F(x)$. Choose $\mathcal{S} \subseteq [n]$ with $B$ samples and $\mathcal{S}_H \subseteq [n]$ with $B_H$ samples, adopt $\eta = \frac{1}{2L(1-\alpha_1\beta_1)}$, where $\alpha_1 \in [0,1]$ and $\beta_1 \in (0,1)$, and assume that $\mathfrak{K}$, $\alpha_1$, $\beta_1$ are further selected so that*

$$\hat{\rho} = \frac{4L\delta(1-\alpha_1\beta_1)}{\mathfrak{K}} < 1. \tag{25}$$

*Then SN-GAG (Algorithm 1) converges linearly in expectation with rate $\hat{\rho}$:*

$$\mathbb{E}\left[\|\nabla F(\tilde{x}^{\mathfrak{S}})\|^2\right] \leq \hat{\rho}^{\mathfrak{S}} \|\nabla F(\tilde{x}^0)\|^2. \tag{26}$$

*Proof.* Technical proofs are available in Appendix B.2. □

Similarly, we easily obtain that the gradient complexity of SN-GAG (Algorithm 1) for the PŁ case is

$$O\left((n + (8B + 4B_H)L\delta(1-\alpha_1\beta_1))\log\left(\frac{1}{\varepsilon}\right)\right). \tag{27}$$

To the best of our knowledge, there are quite less studies to discuss the performance of $\mathcal{SSO}$ methods for the model (2) with the PŁ constraint. In contrast, for $\mathcal{FSO}$ methods, Reddi et al. (2016) proved that SVRG required

$$O\left(\left(n + n^{2/3}\delta\right)\log\left(\frac{1}{\varepsilon}\right)\right) \tag{28}$$

gradient evaluations to acquire an $\varepsilon$-approximate stationary point. In addition, Nguyen et al. (2017b) showed that the overall complexity of SARAH for solving the PŁ case is

$$O\left(\left(n + L^2\delta^2\right)\log\left(\frac{1}{\varepsilon}\right)\right). \tag{29}$$

The results among (27), (28), and (29) demonstrate that with appropriate $B$, $B_H$, $\alpha_1$, and $\beta_1$, SN-GAG (Algorithm 1) attain a lower computational complexity than advanced $\mathcal{FSO}$ methods for the PŁ case.

## 4 STOCHASTIC QUASI-NEWTON ALGORITHMS WITH GENERAL ADAPTIVE GRADIENT

This section develops our second adaptive $\mathcal{SSO}$ method, SQN-NAG. Like the above section, we describe our SQN-GAG method in subsection 4.1. Further, the main theoretical results of SQN-GAG for the SC case and the PŁ case are established in subsection 4.2.

### 4.1 SQN-GAG

The most broadly employed stochastic version of quasi-Newton method, the BFGS-like method (Byrd et al., 2016; Wang et al., 2017; Zhang et al., 2021), renews $B_k$ via

$$B_k = B_{k-1} + \frac{y_{k-1}y_{k-1}^T}{s_{k-1}^T y_{k-1}} - \frac{B_{k-1}s_{k-1}s_{k-1}^T B_{k-1}}{s_{k-1}^T B_{k_1} s_{k-1}}, \tag{30}$$

where $s_{k-1} = x_k - x_{k-1}$ and $y_{k-1} = \nabla F_{\mathcal{S}}(x_k) - \nabla F_{\mathcal{S}}(x_{k-1})$. By virtue of utilizing the Sherman-Morrison-Woodbury formula, one derives that the equivalent update to $H_k = B_k^{-1}$ is

$$H_k = (I - \rho_{k-1}s_{k-1}y_{k-1}^T)H_{k-1}(I - \rho_{k-1}y_{k-1}s_{k-1}^T) + \rho_{k-1}s_{k-1}s_{k-1}^T, \tag{31}$$

---

**Algorithm 2** SQN-GAG

---

**Require:** base learning rate $\eta$, outer loop size $\mathfrak{S}$, inner loop size $\mathfrak{K}$, batch sample $B$, the preconditioned parameters $\epsilon$, $\alpha_1$, $\alpha_2$, $\beta_1$, and $\beta_2$

    **Initialize:** $\widetilde{x}^0$, $G_0^0 = \mathbf{0}$, $U_0^0 = \mathbf{0}$, and $H_0 = I_d$

    **for** $s = 1$ **to** $\mathfrak{S}$ **do**

        $\tilde{x} = x_0^s = \widetilde{x}^{s-1}$

        $g^s = \nabla F(\tilde{x})$

        **for** $k = 1$ **to** $\mathfrak{K}$ **do**

            Select a sample $\mathcal{S} \subset [n]$, with $\mathcal{S} = B$, and Compute first-order stochastic gradient estimator, $V_k^s$

        $V_k^s = \nabla F_\mathcal{S}(x_{k-1}^s) - \nabla F_\mathcal{S}(\tilde{x}) + g^s$

            Select $\mathcal{S}_H \subset [n]$, with $|\mathcal{S}_H| = B_H$, and update the Newton direction by using the approximate information of Hessian matrix, $H_k^s \approx (\nabla^2 F_{S_H}(x_k^s))^{-1}$ (defined by equation 31)

            **if** $k < 1$ **then**

                $\bar{V}_{\mathrm{ND}}^s = V_k^s$

            **else**

                $\bar{V}_{\mathrm{ND}}^s = H_k^s V_k^s$

            **end if**

            Compute momentum

        $G_k^s = \beta_1 G_{k-1}^s + (1 - \beta_1)\bar{V}_{\mathrm{ND}}^s$

            Compute adaptive learning rate

        $U_k^s = \beta_2 U_{k-1}^s + (1 - \beta_2)(\nabla F_\mathcal{S}(x_{k-1}^s))^2$

            Update parameters

        $x_k^s = x_{k-1}^s - \eta \cdot \dfrac{\alpha_1 G_k^s + (1-\alpha_1)\bar{V}_{\mathrm{ND}}^s}{\sqrt{\alpha_2 U_k^s + (1-\alpha_2)(\nabla F_\mathcal{S}(x_{k-1}^s))^2} + \epsilon}$

        **end for**

        $\widetilde{x}^s = x_{\mathfrak{K}}^s$

    **end for**

---

where $\rho_{k-1} = 1/(s_{k-1}^T y_{k-1})$.

We show the algorithmic framework of our SQN-GAG method in Algorithm 2.

**Remark 2:** Review SN-GAG (Algorithm 1) carefully, we easily observe that the only difference between SN-GAG (Algorithm 1) and SQN-GAG (Algorithm 2) is that the latter uses an approximation of real Hessian matrix. More generally, it is easy to find the connections between SQN-GAG (Algorithm 2) and the most of existing adaptive gradient methods. To avoid redundancy, we, here, do not show more details of the relationships between SQN-GAG (Algorithm 2) and the existing adaptive gradient methods. Actually, we can also incorporate GAG into the L-BFGS framework to obtain another novel $\mathcal{SSO}$ method.

### 4.2 Convergence Analysis for SQN-GAG

In order to finish the proof of SQN-GAG (Algorithm 2), the following lemmas are required.

According to the work (Byrd et al., 2016; Chen & Feng, 2023), we have the following result about the Hessian approximate generated by SQN-GAG (Algorithm 2)

**Lemma 1.** *Let Assumption 1 and Assumption 2 hold. There exist constants $0 < M_1 \leq M_2$ such that for $k = 1, 2, \ldots,$ the Hessian approximations $\{H_k^s\}$ resulting from SQN-GAG (Algorithm 2) satisfy the inequality:*

$$M_1 I \prec H_k^s \prec M_2 I. \tag{32}$$

Following, Lemma 2 establishes the bound of $\bar{V}_{\mathrm{ND}}^s$ in SQN-GAG (Algorithm 2) .

**Lemma 2.** *Suppose the Assumption 1(c) and Lemma 1 hold, and $\theta > 0$. We have the bound of the update direction, $\bar{V}_{\mathrm{ND}}^s$, shown in SQN-GAG (Algorithm 2),*

$$\|\bar{V}_{\mathrm{ND}}^s\|^2 \geq \left(2\theta M_1 - \frac{\theta^2}{2} - M_1^2\right)\|\nabla F(x_{k-1}^s)\|^2 - \left(2\theta M_1 - \frac{\theta^2}{2} - M_1^2\right)\frac{4\sigma^2}{B}. \tag{33}$$

*Proof.* The proof of Lemma 2 can follow from that of Lemma 3.   □

### 4.2.1 CONVERGENCE PROPERTY ON THE SC CASE

Theorem 3 provides the main theoretical result of SNG-GAG (Algorithm 2) for the SC case.

**Theorem 3.** *Let model (2) satisfy Assumption 1(a), Assumption 1(c), Assumption 3 and $x_* = \arg\min_{x \in \mathbb{R}^d} F(x)$. Choose $\mathcal{S} \subseteq [n]$ with $B$ samples and $\mathcal{S}_H \subseteq [n]$ with $B_H$ samples, set $\eta = \frac{1}{2L(1-\alpha_1\beta_1)}$, where $\alpha_1 \in [0,1]$ and $\beta_1 \in (0,1)$ and assume that $\mathfrak{K}$, $\alpha_1$, $\beta_1$ are further chosen so that*

$$\rho = \frac{2L(1-\alpha_1\beta_1)}{\mu\mathfrak{K}} < 1. \tag{34}$$

*Then SQN-GAG (Algorithm 2) also attains a linear convergence speed in expectation with rate $\rho$:*

$$\mathbb{E}\left[\|\nabla F(\tilde{x}^{\mathfrak{S}})\|^2\right] \leq \rho^{\mathfrak{S}}\|\nabla F(\tilde{x}^0)\|^2. \tag{35}$$

*Additionally, the complexity of SQN-GAG (Algorithm 2) for the SC objective function is*

$$O\left((n + 2\kappa(2B + 2B_H)(1-\alpha_1\beta_1))\log\left(\frac{1}{\varepsilon}\right)\right). \tag{36}$$

### 4.2.2 CONVERGENCE PROPERTY ON THE PŁ CASE

The main theoretical results of SQN-GAG (Algorithm 2) for the PŁ case are provided in Theorem 4.

**Theorem 4.** *Let model (2) satisfy Assumptions 1-2 and $x_* = \arg\min_{x \in \mathbb{R}^d} F(x)$. Choose $\mathcal{S} \subseteq [n]$ with $B$ samples and $\mathcal{S}_H \subseteq [n]$ with $B_H$ samples, set $\eta = \frac{1}{2L(1-\alpha_1\beta_1)}$, where $\alpha_1 \in [0,1]$ and $\beta_1 \in (0,1)$ and assume that $\mathfrak{K}$, $\alpha_1$, $\beta_1$ are further selected so that*

$$\hat{\rho} = \frac{4L\delta(1-\alpha_1\beta_1)}{\mathfrak{K}} < 1. \tag{37}$$

*Then SQN-GAG (Algorithm 2) attains a linear convergence speed in expectation with rate $\hat{\rho}$:*

$$\mathbb{E}\left[\|\nabla F(\tilde{x}^{\mathfrak{S}})\|^2\right] \leq \hat{\rho}^{\mathfrak{S}}\|\nabla F(\tilde{x}^0)\|^2. \tag{38}$$

*Moreover, the complexity of SQN-GAG (Algorithm 2) for the PŁ objective function is*

$$O\left((n + 4L\delta\kappa(2B + 2B_H)(1-\alpha_1\beta_1))\log\left(\frac{1}{\varepsilon}\right)\right). \tag{39}$$

The results in Theorem 3 and Theorem 4 imply that SQN-GAG (Algorithm 2) have similar theoretical properties to SN-GAG (Algorithm 1). This is the indeed case, since SN-GAG (Algorithm 1) and SQN-GAG (Algorithm 2) adopt the similar algorithmic framework, apart from the way of acquiring Hessian matrix. More specifically, the proof of Theorem 3 and Theorem 4 can easily follow from the proof of Theorem 1 and Theorem 2, respectively.

## 5 CONCLUSION

Motivated by the gap between the $\mathcal{SSO}$ methods and the learning rate, this work developed a class of adaptive $\mathcal{SSO}$ methods from the perspective of adaptive gradient methods. More generally, we proposed to use a GAG method, encompassing most of existing adaptive gradient method (e.g., Adam, AdaGrad, RMSProp, etc.), to automatically compute the learning rate for $\mathcal{SSO}$ methods. Specifically, we applied GAG into two stochastic versions of Newton-like methods, SN and SQN, leading to two new methods: SN-GAG (Algorithm 1) and SQN-GAG (Algorithm 2). We theoretically understood the role of GAG in $\mathcal{SSO}$ methods. Concretely, we proved that SN-GAG (Algorithm 1) and SQN-GAG (Algorithm 2) converged linearly under different backgrounds (a.k.a. the SC case and the PŁ case) and showed that the computational complexities of the resulting $\mathcal{SSO}$ methods were comparable to state-of-the-art $\mathcal{FSO}$ methods and $\mathcal{SSO}$ methods, where the property of $\mathcal{SSO}$ methods under the PŁ case was lacking in existing literature. The applications of SN-GAG (Algorithm 1) and SQN-GAG (Algorithm 2) in different machine learning tasks confirmed their efficacy by comparing with adaptive gradient methods, first-order stochastic methods and second-order stochastic methods. Further, a large number of numerical experiments demonstrated the robustness of the resulting algorithms.

**Statement.** The research conducted in the paper conform, in every respect, with the ICLR Code of Ethics https://iclr.cc/public/CodeOfEthics.

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

# A  APPENDIX

## A.1  NUMERICAL EXPERIMENTS

This section will present an experimental evaluation for SN-GAG (Algorithm 1) and SQN-GAG (Algorithm 2). We first numerically compare the convergence properties of SN-GAG (Algorithm 1) and SQN-GAG (Algorithm 2) with classical adaptive gradient methods, state-of-the-art SSO methods and first-order stochastic methods for strongly-convex and non-convex optimization problems respectively in subsection A.3. Further, we investigate the effect of several crucial parameters in both SN-GAG (Algorithm 1) and SQN-GAG (Algorithm 2) in subsection A.4.

## A.2  EXPERIMENTAL DETAILS

Two common machine learning tasks, the logistic regression (LR) model with the $\ell_2$ regularization term and the non-convex squared hinge loss support vector machine (SVM) model with the $\ell_2$ regularization term, are used, i.e.,

$$(\textbf{LR}) \quad \min_{x \in \mathbb{R}^d} F(x) = \frac{1}{n} \sum_{i=1}^{n} \log(1 + \exp(-b_i a_i^T x)) + \frac{\lambda}{2} \|x\|^2, \tag{40}$$

$$(\textbf{SVM}) \quad \min_{x \in \mathbb{R}^d} F(x) = \frac{1}{n} \sum_{i=1}^{n} \left( \left[ 1 - b_i a_i^T x \right]_+ \right)^2 + \frac{\lambda}{2} \|x\|^2, \tag{41}$$

where $a_i \in \mathbb{R}^d$ is the $i$th data point and $b_i \in \pm 1$ denotes the corresponding label. In our experiments, we adopt the value of $\lambda = 10^{-2}$.

We implement all numerical experiments on four public datasets from the LIBSVM Chang & Lin (2011), where these datasets, $a8a$, $ijcnn1$, $covtype$, and $w8a$, are summarized in Table 1. Without otherwise specified, in all figures, the $x$-axis represents the number of effective passes and the $y$-axis denotes the objective gap, $F(\tilde{x}^s) - F(x_*)$.

Table 1: Descriptions of DataSets

| Dataset | Sample size (n) | Dimension (d) |
|---------|-----------------|---------------|
| $a8a$ | 22,696 | 123 |
| $covtype$ | 581,012 | 54 |
| $ijcnn1$ | 49,990 | 22 |
| $w8a$ | 49,749 | 300 |

## A.3  COMPARISON WITH OTHER RELATED METHODS

We compare SN-GAG (Algorithm 1) and SQN-GAG (Algorithm 2) with the related methods, containing AdaGrad Duchi et al. (2011), Adam Kingma (2014), RMSProp Tieleman et al. (2012), AMSGRAD Reddi et al. (2018), SVRG Johnson & Zhang (2013), SARAH Nguyen et al. (2017a), and SLBFGS Mokhtari & Ribeiro (2020). AdaGrad, Adam, RMSProp, and AMSGRAD are four classical adaptive gradient methods. Specifically, AMSGRAD is a variant of Adam, depending on long-term memory of historical gradients and converging with higher probability than Adam. SVRG and SARAH are two popular first-order stochastic methods and converge linearly for the SC objective function. SLBFGS introduces the SVRG-like gradient estimator into LBFGS.

The parameter settings for different comparative methods are described here. When executing RMSProp, the parameter $\beta$ is set to be $\beta = 0.9$. We perform Adam and AMSGRAD with $\beta_1 = 0.9$ and $\beta_2 = 0.999$ for different datasets. SVRG, SARAH and SLBFGS work with a constant learning rate, where we select the learning rate for SVRG, SARAH, and SLBFGS from multiple learning rates which make them behave better. When executing SN-GAG (Algorithm 1) and SQN-GAG (Algorithm 2), we set $\alpha_1 = 0.9$, $\alpha_2 = 0.9$, $\beta_1 = 0.9$, $\beta_2 = 0.9$, and $\epsilon = 1$ for all datasets.

We show the experimental results among these methods on LR and SVM models on four datasets in Fig. 1. For clarity, the performance of different methods on the LR model and the SVM model

is reported in the first line of Fig. 1 and the second line of Fig. 1, respectively. Fig. 1 demonstrates that SN-GAG (Algorithm 1) and SQN-GAG (Algorithm 2) converges linearly on different datasets. Moreover, Fig. 1 confirms that for different models, on almost all datasets, SN-GAG (Algorithm 1) and SQN-GAG (Algorithm 2) attain the fastest convergence rate than state-of-the-art stochastic methods. In addition, the comparison results among SN-GAG (Algorithm 1) and SQN-GAG (Algorithm 2), SLBFGS, and the original SVRG method validate the positive role of second-order information in modifying first-order stochastic methods.

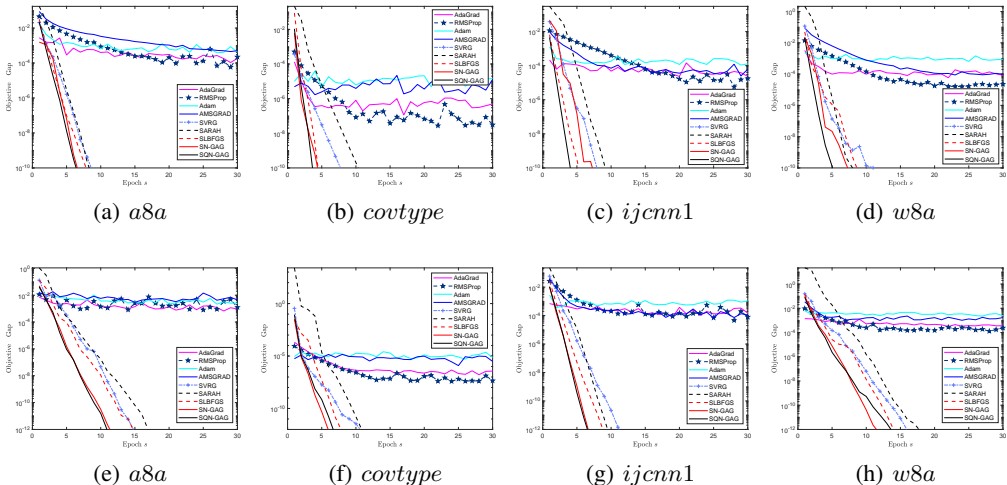

(a) $a8a$     (b) $covtype$     (c) $ijcnn1$     (d) $w8a$

(e) $a8a$     (f) $covtype$     (g) $ijcnn1$     (h) $w8a$

Figure 1: First row: performance comparison for solving the LR model among different methods on $a8a$, $covtype$, $ijcnn1$, and $w8a$. Second row: performance comparison for solving the SVM model among different methods on $a8a$, $covtype$, $ijcnn1$, and $w8a$.

### A.4 THE EFFECT OF DIFFERENT HYPER-PARAMETERS

The exploration of the resulting adaptive SSO methods with different hyper-parameters is discussed here. Note that, for convenience but without loss generality, in the following, we will perform SN-GAG (Algorithm 1) in the first line of all figures and perform SQN-GAG (Algorithm 2) in the second line of all figures. Moreover, we fastened other hyper-parameters when discussing the impact of one of these hyper-parameters in the resulting algorithms.

**Effect of $\beta_1$.** We start this part from investing the effect of $\beta_1$ in SN-GAG (Algorithm 1) and SQN-GAG (Algorithm 2) by performing them on different datasets and show the results in Fig. 2. The behavior of SN-GAG (Algorithm 1) and SQN-GAG (Algorithm 2) we consider is under the case that $\beta_1$ is chosen from $\{0.1, 0.3, 0.5, 0.7, 0.9\}$. For other parameters, we set $\alpha_1 = 0.9$, $\alpha_2 = 0.9$, $\beta_2 = 0.9$ on all datasets. As Fig. 2 shows, both SN-GAG (Algorithm 1) and SQN-GAG (Algorithm 2) are insensitive to $\beta_1$ on different datasets.

**Effect of $\beta_2$.** Further, Fig. 3 explores the performance of SN-GAG (Algorithm 1) and SQN-GAG (Algorithm 2) with different $\beta_2$, where $\beta_2$ is also selected from the set $\{0.1, 0.3, 0.5, 0.7, 0.9\}$. For other parameters, we adopt $\alpha_1 = 0.9$, $\alpha_2 = 0.9$, and $\beta_1 = 0.4$ when performing the resulting methods on different datasets. Obviously, Fig. 3 demonstrates the robustness of SN-GAG (Algorithm 1) and SQN-GAG (Algorithm 2) to $\beta_2$.

**Effect of $\alpha_1$.** Fig. 4 explores how the parameter $\alpha_1$ influences SN-GAG (Algorithm 1) and SQN-GAG (Algorithm 2). The parameter $\alpha_1$ is considered in $\{0, 0.2, 0.4, 0.6, 0.8, 1\}$. The other parameters are fixed to be $\beta_1 = 0.9$, $\beta_2 = 0.9$, $\alpha_2 = 0.9$, respectively. The results in Fig. 4 imply the insensitivity of SN-GAG (Algorithm 1) and SQN-GAG (Algorithm 2) to $\alpha_1$.

**Effect of $\alpha_2$.** Finally, in Fig. 5, we show numerical properties of the resulting methods with different $\alpha_2$, where $\alpha_2$ is considered in $\{0, 0.2, 0.4, 0.6, 0.8, 1\}$ as well. The other parameters are set to be $\beta_1 = 0.9$, $\beta_2 = 0.9$, $\alpha_1 = 0.9$ for different datasets. Obviously, Fig. 5 demonstrates that SN-GAG (Algorithm 1) and SQN-GAG (Algorithm 2) are insensitive to $\alpha_2$.

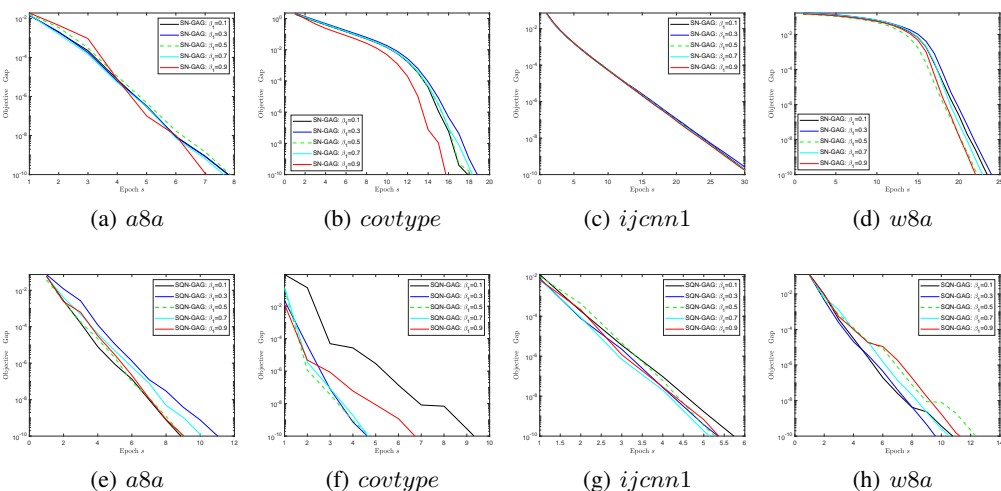

Figure 2: First row: performance comparison for addressing LR with different selections of $\beta_1$ in SN-GAG (Algorithm 1) on $a8a$, $covtype$, $ijcnn1$, and $w8a$. Second row: performance comparison for addressing SVM with different selections of $\beta_1$ in SQN-GAG (Algorithm 2) on $a8a$, $covtype$, $ijcnn1$, and $w8a$. Specifically, we select the hyper-parameter $\beta_1$ from $\{0.1, 0.3, 0.5, 0.7, 0.9\}$.

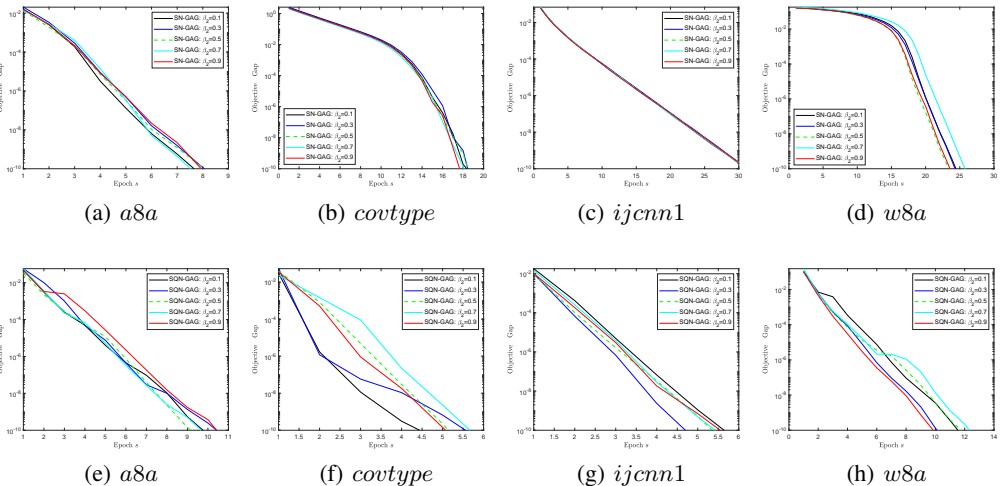

Figure 3: First row: performance comparison for addressing LR with different selections of $\beta_2$ in SN-GAG (Algorithm 1) on $a8a$, $covtype$, $ijcnn1$, and $w8a$. Second row: performance comparison for addressing SVM with different selections of $\beta_2$ in SQN-GAG (Algorithm 2) on $a8a$, $covtype$, $ijcnn1$, and $w8a$. Specifically, we select the hyper-parameter $\beta_2$ from $\{0.1, 0.3, 0.5, 0.7, 0.9\}$ as well.

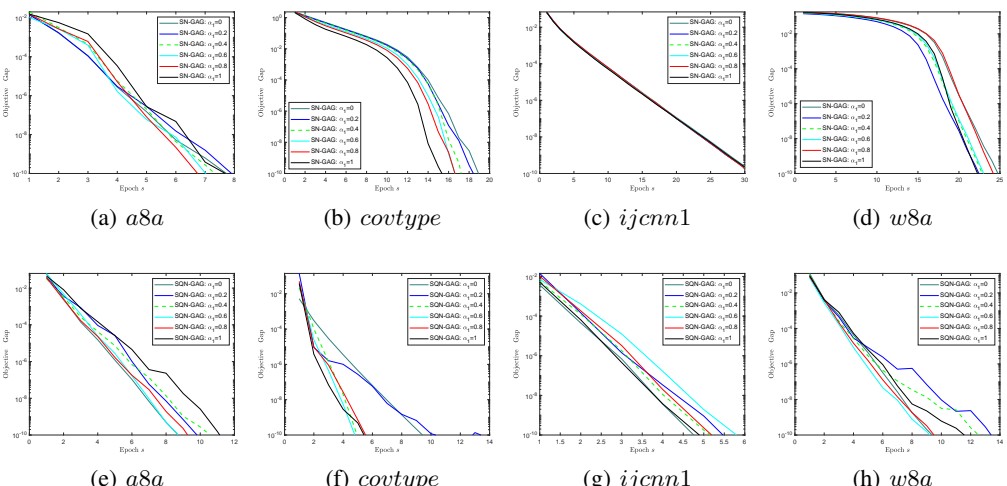

Figure 4: First row: performance comparison for addressing LR with different selections of $\alpha_1$ in SN-GAG (Algorithm 1) on $a8a$, $covtype$, $ijcnn1$, and $w8a$. Second row: performance comparison for addressing SVM with different selections of $\alpha_1$ in SQN-GAG (Algorithm 2) on $a8a$, $covtype$, $ijcnn1$, and $w8a$. Specifically, we select the hyper-parameter $\alpha_1$ from $\{0, 0.2, 0.4, 0.6, 0.8, 1\}$.

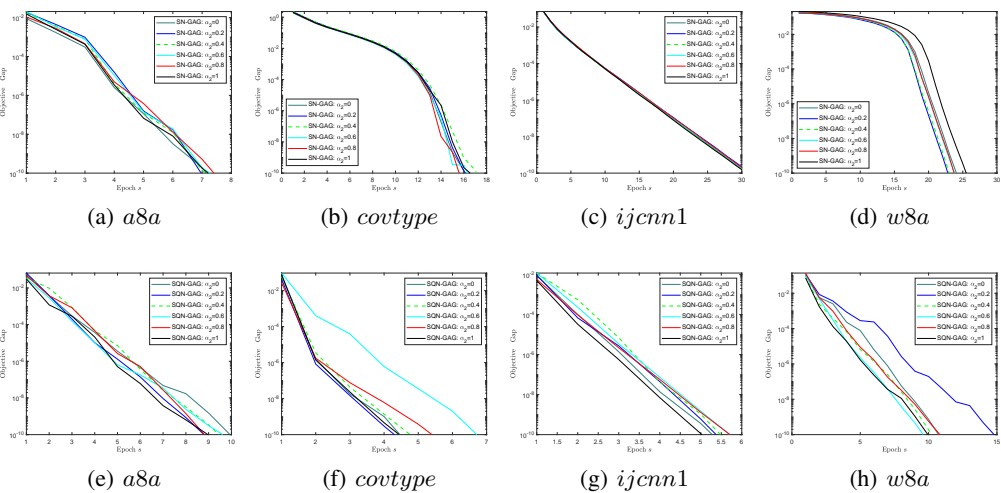

Figure 5: First row: performance comparison for addressing LR with different selections of $\alpha_2$ in SN-GAG (Algorithm 1) on $a8a$, $covtype$, $ijcnn1$, and $w8a$. Second row: performance comparison for addressing SVM with different selections of $\alpha_2$ in SQN-GAG (Algorithm 2) on $a8a$, $covtype$, $ijcnn1$, and $w8a$. Specifically, we select the hyper-parameter $\alpha_2$ from $\{0, 0.2, 0.4, 0.6, 0.8, 1\}$.

Summarily, all results in Fig. 2, Fig. 3, Fig. 4, and Fig. 5 confirm the robustness of our methods, SN-GAG (Algorithm 1) and SQN-GAG (Algorithm 2), to crucial hyper-parameters, which significantly reduce the difficulty of the practitioners in setting these key hyper-parameters.

# B PROOFS FOR SN-GAG

We begin this part with some useful lemmas for SN-GAG (Algorithm 1) below.

**Lemma 3.** *Suppose the Assumption 1(c) holds. We obtain the bound of the update direction, $V_{\mathrm{ND}}^s$, shown in SN-GAG (Algorithm 1), i.e.,*

$$\|V_{\mathrm{ND}}^s\|^2 \geq \left(\frac{\theta^2}{4} + \frac{1}{2\Lambda} - \frac{\theta}{2}\right) \|\nabla F(x_{k-1}^s)\|^2 - \left(\frac{\theta^2}{2} + \frac{1}{\Lambda} - \theta\right)\frac{4\sigma^2}{B}, \tag{42}$$

*where $\theta > 0$.*

*Proof.* The fact $\|x\|^2 \geq \frac{1}{2}\|y\|^2 - \|y - x\|^2$ and the definition $V_{\mathrm{ND}}^s = (\nabla^2 F_{\mathcal{S}_H}(x_k^s))^{-1}V_k^s$ in SN-GAG (Algorithm 1) ensure

$$\|V_{\mathrm{ND}}^s\|^2 = \frac{1}{2}\|\theta V_k^s\|^2 - \|\theta V_k^s - (\nabla^2 F_{\mathcal{S}_H}(x_k^s))^{-1}V_k^s\|^2$$

$$= \frac{1}{2}\|\theta V_k^s\|^2 - \|(\theta I - (\nabla^2 F_{\mathcal{S}_H}(x_k^s))^{-1})V_k^s\|^2$$

$$\geq \frac{\theta^2}{2}\|V_k^s\|^2 - \|V_k^s\|^2\|\theta I - (\nabla^2 F_{\mathcal{S}_H}(x_k^s))^{-1}\|^2$$

$$= \|V_k^s\|^2\left(\frac{\theta^2}{2} - \|\theta I - (\nabla^2 F_{\mathcal{S}_H}(x_k^s))^{-1}\|^2\right)$$

$$\geq \left(\frac{2\theta}{\Lambda} - \frac{\theta^2}{2} - \frac{1}{\Lambda^2}\right)\|V_k^s\|^2$$

where the first inequality holds due to the Cauchy-Schwarts inequality, $|x^T y| \leq \|x\| \cdot \|y\|$.

Further, combining the definition $V_k^s = \nabla F_{\mathcal{S}}(x_{k-1}^s) - \nabla F_{\mathcal{S}}(\tilde{x}) + \frac{1}{n}\sum_{i=1}^n \nabla f_i(\tilde{x})$ in SN-GAG (Algorithm 1) and $\|x\|^2 \geq \frac{1}{2}\|y\|^2 - \|y - x\|^2$ , we ascertain

$$\|V_{\mathrm{ND}}^s\|^2 \geq \left(\frac{2\theta}{\Lambda} - \frac{\theta^2}{2} - \frac{1}{\Lambda^2}\right)\left(\frac{1}{2}\|\nabla F(x_{k-1}^s)\|^2 - \|\nabla F(x_{k-1}^s)\right.$$

$$\left. - \nabla F_{\mathcal{S}}(x_{k-1}^s) + \nabla F_{\mathcal{S}}(\tilde{x}) - \nabla F(\tilde{x})\|^2\right)$$

$$\geq \left(\frac{2\theta}{\Lambda} - \frac{\theta^2}{2} - \frac{1}{\Lambda^2}\right)\left(\frac{1}{2}\|\nabla F(x_{k-1}^s)\|^2 - \frac{4\sigma^2}{B}\right)$$

$$= \left(\frac{\theta}{\Lambda} - \frac{\theta^2}{4} - \frac{1}{2\Lambda^2}\right)\|\nabla F(x_{k-1}^s)\|^2 - \left(\frac{2\theta}{\Lambda} - \frac{\theta^2}{2} - \frac{1}{\Lambda^2}\right)\frac{4\sigma^2}{B},$$

where the second inequality uses the fact $\|x + y\|^2 \leq 2\|x\|^2 + 2\|y\|^2$ and Assumption 1(c). $\square$

**Lemma 4.** *Suppose Assumption 1(c) holds. For $U_k^s$, defined in SN-GAG (Algorithm 1), we have the following conclusion:*

$$\left\|\sqrt{\alpha_2 U_k^s + (1 - \alpha_2)(\nabla F_{\mathcal{S}}(x_{k-1}^s))^2} + \epsilon\right\|^2 \leq 2\alpha_2(1 - \beta_2^{\mathfrak{K}})\gamma^2 + 2(1 - \alpha_2)\gamma^2 + 2\epsilon^2, \tag{43}$$

*where $\beta_2 \in (0, 1)$ and $\alpha_2 \in [0, 1]$.*

*Proof.* According to SN-GAG (Algorithm 1), we obtain

$$\left\|\sqrt{\alpha_2 U_k^s + (1 - \alpha_2)(\nabla F_{\mathcal{S}}(x_{k-1}^s)^2} + \epsilon\right\|^2 \leq 2\left\|\sqrt{\alpha_2 U_k^s + (1 - \alpha_2)(\nabla F_{\mathcal{S}}(x_{k-1}^s))^2}\right\|^2 + 2\epsilon^2$$

$$\leq 2\|\alpha_2 U_k^s\| + 2\|(1 - \alpha_2)(\nabla F_{\mathcal{S}}(x_{k-1}^s))^2\| + 2\epsilon^2 \tag{44}$$

where the first inequality uses the condition $(a+b)^2 \leq 2a^2 + 2b^2$ and the second inequality employs the triangle inequality $\|x + y\| \leq \|x\| + \|y\|$.

The definition $U_k^s = \beta_2 U_{k-1}^s + (1 - \beta_2)(\nabla F_{\mathcal{S}}(x_{k-1}^s))^2$ in SN-GAG (Algorithm 1) makes us have

$$U_k^s = (1 - \beta_2) \sum_{i=1}^{\mathfrak{K}} \beta_2^{\mathfrak{K}-i} \left( \nabla F_{\mathcal{S}}(x_{k-1}^s) \right)^2 . \tag{45}$$

Combining the results in (44) and (45), we derive

$$2\|\alpha_2 U_k^s\| + 2\|(1-\alpha_2)(\nabla F_{\mathcal{S}}(x_{k-1}^s))^2\| + 2\epsilon^2$$

$$= 2\alpha_2(1-\beta_2) \left\| \sum_{i=1}^{\mathfrak{K}} \beta_2^{\mathfrak{K}-i}(\nabla F_{\mathcal{S}}(x_{k-1}^s))^2 \right\| + 2(1-\alpha_2)\|(\nabla F_{\mathcal{S}}(x_{k-1}^s))^2\| + 2\epsilon^2$$

$$\leq 2\alpha_2(1-\beta_2) \left[ \beta_2^{\mathfrak{K}-1}\|\nabla F_{\mathcal{S}}(x_1^s)\|^2 + \beta_2^{\mathfrak{K}-2}\|\nabla F_{\mathcal{S}}(x_2^s)\|^2 + \cdots + \|\nabla F_{\mathcal{S}}(x_m^s)\|^2 \right] + 2\|(1 - \alpha_2)(\nabla F_{\mathcal{S}}(x_{k-1}^s))^2\| + 2\epsilon^2$$

$$\overset{(11)}{\leq} 2\alpha_2(1-\beta_2)\left( \beta_2^{\mathfrak{K}-1}\gamma^2 + \beta_2^{\mathfrak{K}-2}\gamma^2 + \cdots + \gamma^2 \right) + 2(1-\alpha_2)\gamma^2 + 2\epsilon^2$$

$$\leq 2\alpha_2(1-\beta_2^{\mathfrak{K}})\gamma^2 + 2(1-\alpha_2)\gamma^2 + 2\epsilon^2, \tag{46}$$

where the first inequality keeps due to the triangle inequality and the second inequality holds due to the condition in (11).

$\square$

## B.1 PROOF OF THEOREM 1

Here, we offer the technical proofs of Theorem 1 and Theorem 2.

*Proof.* The $L$-smooth property of the function and the definition, $x_k^s = x_{k-1}^s - \eta\left[ \frac{\alpha_1 G_k^s + (1-\alpha_1)V_{\mathrm{ND}}^s}{\sqrt{\alpha_2 U_k^s + (1-\alpha_2)(\nabla F_{n_k}(x_{k-1}^s))^2} + \epsilon} \right]$ in SN-GAG (Algorithm 1), ensure

$$\mathbb{E}[F(x_k^s)] \leq \mathbb{E}\left[ F(x_{k-1}^s) + \langle \nabla F(x_{k-1}^s), x_k^s - x_{k-1}^s \rangle + \frac{L}{2}\|x_k^s - x_{k-1}^s\|^2 \right]$$

$$= \mathbb{E}\bigg[ F(x_{k-1}^s) - \eta\alpha_1 \bigg\langle \nabla F(x_{k-1}^s),$$

$$\frac{G_k^s}{\sqrt{\alpha_2 U_k^s + (1-\alpha_2)(\nabla F_{\mathcal{S}}(x_{k-1}^s))^2} + \epsilon} \bigg\rangle - \eta(1-\alpha_1) \bigg\langle \nabla F(x_{k-1}^s),$$

$$\frac{V_{\mathrm{ND}}^s}{\sqrt{\alpha_2 U_k^s + (1-\alpha_2)(\nabla F_{\mathcal{S}}(x_{k-1}^s))^2} + \epsilon} \bigg\rangle + \frac{L\eta^2}{2}$$

$$\cdot \left\| \frac{\alpha_1 G_k^s + (1-\alpha_1)V_{\mathrm{ND}}^s}{\sqrt{\alpha_2 U_k^s + (1-\alpha_2)(\nabla F_{\mathcal{S}}(x_{k-1}^s))^2} + \epsilon} \right\|^2 \bigg]. \tag{47}$$

The definition, $G_k^s = \beta_1 G_{k-1}^s + (1 - \beta_1)V_{\mathrm{ND}}^s$, in SN-GAG (Algorithm 1), makes us further obtain

$$F(x_k^s) \leq F(x_{k-1}^s) - \eta\alpha_1\beta_1 \bigg\langle \nabla F(x_{k-1}^s), \frac{G_{k-1}^s}{\sqrt{\alpha_2 U_k^s + (1-\alpha_2)(\nabla F_{\mathcal{S}}(x_{k-1}^s))^2} + \epsilon} \bigg\rangle$$

$$- \eta(1-\alpha_1\beta_1) \bigg\langle \nabla F(x_{k-1}^s), \frac{V_{\mathrm{ND}}^s}{\sqrt{\alpha_2 U_k^s + (1-\alpha_2)(\nabla F_{\mathcal{S}}(x_{k-1}^s))^2} + \epsilon} \bigg\rangle$$

$$+ \frac{L\eta^2}{2} \left\| \frac{\alpha_1\beta_1 G_{k-1}^s + (1-\alpha_1\beta_1)V_{\mathrm{ND}}^s}{\sqrt{\alpha_2 U_k^s + (1-\alpha_2)(\nabla F_{\mathcal{S}}(x_{k-1}^s))^2} + \epsilon} \right\|^2. \tag{48}$$

The use of the facts (i) $\langle a, b \rangle = \frac{1}{2}\left[\|a\|^2 + \|b\|^2 - \|a-b\|^2\right]$ and (ii) $\|a+b\|^2 \leq 2\|a\|^2 + 2\|b\|^2$ further make us ascertain

$$
F(x_k^s) \overset{(i)}{\underset{(ii)}{\leq}} F(x_{k-1}^s) - \frac{\eta\alpha_1\beta_1}{2}\left[\|\nabla F(x_{k-1}^s)\|^2 + \left\|\frac{G_{k-1}^s}{\sqrt{\alpha_2 U_k^s + (1-\alpha_2)(\nabla F_{\mathcal{S}}(x_{k-1}^s))^2} + \epsilon}\right\|^2\right.
$$

$$
\left. - \left\|\nabla F(x_{k-1}^s) - \frac{G_{k-1}^s}{\sqrt{\alpha_2 U_k^s + (1-\alpha_2)(\nabla F_{\mathcal{S}}(x_{k-1}^s))^2} + \epsilon}\right\|^2\right] - \frac{\eta(1-\alpha_1\beta_1)}{2}\left[\|\nabla F(x_{k-1}^s)\|^2\right.
$$

$$
\left. + \left\|\frac{V_{\mathrm{ND}}^s}{\sqrt{\alpha_2 U_k^s + (1-\alpha_2)(\nabla F_{\mathcal{S}}(x_{k-1}^s))^2} + \epsilon}\right\| - \left\|\nabla F(x_{k-1}^s) - \frac{V_{\mathrm{ND}}^s}{\sqrt{\alpha_2 U_k^s + (1-\alpha_2)(\nabla F_{\mathcal{S}}(x_{k-1}^s))^2} + \epsilon}\right\|^2\right]
$$

$$
+ \frac{L\eta^2\alpha_1^2\beta_1^2\|G_{k-1}^s\|^2 + L\eta^2(1-\alpha_1\beta_1)^2\|V_{\mathrm{ND}}^s\|^2}{\left\|\sqrt{\alpha_2 U_k^s + (1-\alpha_2)(\nabla F_{\mathcal{S}}(x_{k-1}^s))^2} + \epsilon\right\|^2}
$$

$$
= F(x_{k-1}^s) - \frac{\eta}{2}\|\nabla F(x_{k-1})^s\|^2 - \frac{\eta\alpha_1\beta_1}{2}\cdot\left\|\frac{G_{k-1}^s}{\sqrt{\alpha_2 U_k^s + (1-\alpha_2)(\nabla F_{\mathcal{S}}(x_{k-1}^s))^2} + \epsilon}\right\|^2 + \frac{\eta\alpha_1\beta_1}{2}
$$

$$
\cdot\left\|\nabla F(x_{k-1}^s) - \frac{G_{k-1}^s}{\sqrt{\alpha_2 U_k^s + (1-\alpha_2)(\nabla F_{\mathcal{S}}(x_{k-1}^s))^2} + \epsilon}\right\|^2
$$

$$
- \frac{\eta(1-\alpha_1\beta_1)}{2}\left\|\frac{V_{\mathrm{ND}}^s}{\sqrt{\alpha_2 U_k^s + (1-\alpha_2)(\nabla F_{\mathcal{S}}(x_{k-1}^s))^2} + \epsilon}\right\|^2 + \frac{\eta(1-\alpha_1\beta_1)}{2}\left\|\nabla F(x_{k-1}^s)\right.
$$

$$
\left. - \frac{V_{\mathrm{ND}}^s}{\sqrt{\alpha_2 U_k^s + (1-\alpha_2)(\nabla F_{\mathcal{S}}(x_{k-1}^s))^2} + \epsilon}\right\|^2 + \frac{L\eta^2\alpha_1^2\beta_1^2\|G_{k-1}^s\|^2 + L\eta^2(1-\alpha_1\beta_1)^2\|V_{\mathrm{ND}}^s\|^2}{\left\|\sqrt{\alpha_2 U_k^s + (1-\alpha_2)(\nabla F_{\mathcal{S}}(x_{k-1}^s))^2} + \epsilon\right\|^2}.
$$

$$\tag{49}$$

To satisfy the inequality (49), it is enough to keep the following condition

$$
F(x_k^s) \leq F(x_{k-1}^s) - \frac{\eta}{2}\|\nabla F(x_{k-1}^s)\|^2 - \left[\frac{\eta(1-\alpha_1\beta_1)}{2} - L\eta^2(1-\alpha_1\beta_1)^2\right]
$$

$$
\cdot\left\|\frac{V_{\mathrm{ND}}^s}{\sqrt{\alpha_2 U_k^s + (1-\alpha_2)(\nabla F_{\mathcal{S}}(x_{k-1}^s))^2} + \epsilon}\right\|^2.
$$

$$\tag{50}$$

Considering the results in Lemma 3 and Lemma 4, we have

$$
F(x_k^s) \overset{Lemma\ 4}{\leq} F(x_{k-1}^s) - \frac{\eta}{2}\|\nabla F(x_{k-1}^s)\|^2 - \left[\frac{\eta(1-\alpha_1\beta_1)}{2}\right.
$$

$$
\left. - L\eta^2(1-\alpha_1\beta_1)^2\right]\frac{\|V_{\mathrm{ND}}^s\|^2}{2\alpha_2(1-\beta_2^m)\gamma^2 + 2(1-\alpha_2)\gamma^2 + 2\epsilon^2}
$$

$$
\overset{Lemma\ 3}{\leq} F(x_{k-1}^s) - \frac{\eta}{2}\|\nabla F(x_{k-1}^s)\|^2
$$

$$
- \frac{\eta(1-\alpha_1\beta_1) - 2L\eta^2(1-\alpha_1\beta_1)}{4\alpha_2(1-\beta_2^{\mathfrak{K}})\gamma^2 + 4(1-\alpha_2)\gamma^2 + 4\epsilon^2}\left[\left(\frac{\theta}{\Lambda} - \frac{\theta^2}{4} - \frac{1}{2\Lambda^2}\right)\right.
$$

$$
\left. \cdot \|\nabla F(x_{k-1}^s)\|^2 - \left(\frac{2\theta}{\Lambda} - \frac{\theta^2}{2} - \frac{1}{\Lambda^2}\right)\frac{4\sigma^2}{B}\right]
$$

$$
= F(x_{k-1}^s) - \left[\frac{\eta}{2} + \frac{\eta(1-\alpha_1\beta_1) - 2L\eta^2(1-\alpha_1\beta_1)^2}{4\alpha_2(1-\beta_2^m)\gamma^2 + 4(1-\alpha_2)\gamma^2 + 4\epsilon^2}\right.
$$

$$
\left. \cdot \left(\frac{\theta}{\Lambda} - \frac{\theta^2}{4} - \frac{1}{2\Lambda^2}\right)\right]\|\nabla F(x_{k-1}^s)\|^2 + \left(\frac{\theta}{\Lambda} + \frac{\theta^2}{2} - \frac{1}{\Lambda^2}\right)\frac{4\sigma^2}{B}
$$

$$
\cdot \frac{\eta(1-\alpha_1\beta_1) - 2L\eta^2(1-\alpha_1\beta_1)^2}{4\alpha_2(1-\beta_2^{\mathfrak{K}})\gamma^2 + 4(1-\alpha_2)\gamma^2 + 4\epsilon^2}. \tag{51}
$$

Telescoping the inequality (51) over $k = 1, \cdots, \mathfrak{K}$, we have

$$
F(x_k^s) \leq F(x_0^s) - \left[\frac{\eta}{2} + \frac{\eta(1-\alpha_1\beta_1) - 2L\eta^2(1-\alpha_1\beta_1)^2}{4\alpha_2(1-\beta_2^{\mathfrak{K}})\gamma^2 + 4(1-\alpha_2)\gamma^2 + 4\epsilon^2}\right.
$$

$$
\left. \cdot \left(\frac{\theta}{\Lambda} - \frac{\theta^2}{4} - \frac{1}{2\Lambda^2}\right)\right]\sum_{k=1}^{\mathfrak{K}}\|\nabla F(x_{k-1}^s)\|^2 + \left(\frac{2\theta}{\Lambda} - \frac{\theta^2}{2} - \frac{1}{\Lambda^2}\right)\frac{4\sigma^2\mathfrak{K}}{B}
$$

$$
\cdot \frac{\eta(1-\alpha_1\beta_1) - 2L\eta^2(1-\alpha_1\beta_1)^2}{4\alpha_2(1-\beta_2^{\mathfrak{K}})\gamma^2 + 4(1-\alpha_2)\gamma^2 + 4\epsilon^2}. \tag{52}
$$

Rearranging the inequality (52), we further have the following inequality

$$
\sum_{k=1}^{\mathfrak{K}}\|\nabla F(x_{k-1}^s)\|^2 \leq \frac{[4\alpha_2(1-\beta_2^{\mathfrak{K}})\gamma^2 + 4(1-\alpha_2)\gamma^2 + 4\epsilon^2]4\Lambda^2}{\mathcal{R}}[F(x_0^s) - F(x_*)]
$$

$$
+ \frac{[\eta(1-\alpha_1\beta_1) - 2L\eta^2(1-\alpha_1\beta_1)^2](4\theta\Lambda - \theta^2\Lambda^2 - 2)8\mathfrak{K}\sigma^2}{B\mathcal{R}}, \tag{53}
$$

where we set $\mathcal{R} = 2\eta\Lambda^2[4\alpha_2(1-\beta_2^{\mathfrak{K}})\gamma^2 + 4(a-\alpha_2)\gamma^2 + 4\varepsilon^2] + [\eta(1-\alpha_1\beta_1) - 2L\eta^2(1-\alpha_1\beta_1)^2](4\theta\Lambda - \theta^2\Lambda^2 - 2)$. Additionally, the above inequality also uses the fact $x_* = \arg\min F(x)$.

Since $\mathbb{E}[\|\nabla F(x_{\mathfrak{K}}^s)\|^2] = \frac{1}{\mathfrak{K}}\sum_{k=1}^{\mathfrak{K}}\|\nabla F(x_k^s)\|^2$, we have

$$
\mathbb{E}[\|\nabla F(x_{\mathfrak{K}}^s)\|^2] \leq \frac{[4\alpha_2(1-\beta_2^{\mathfrak{K}})\gamma^2 + 4(1-\alpha_2)\gamma^2 + 4\epsilon^2]4\Lambda^2}{\mathcal{R}\mathfrak{K}}[F(x_0^s) - F(x_*)]
$$

$$
+ \frac{[\eta(1-\alpha_1\beta_1) - 2L\eta^2(1-\alpha_1\beta_1)^2](4\theta\Lambda - \theta^2\Lambda^2 - 2)8\sigma^2}{B\mathcal{R}}. \tag{54}
$$

According to the SC property of the function (a.k.a. Assumption 3), we easily have the following result

$$
\mathbb{E}[\|\nabla F(x_{\mathfrak{K}}^s)\|^2] \leq \frac{[4\alpha_2(1-\beta_2^{\mathfrak{K}})\gamma^2 + 4(1-\alpha_2)\gamma^2 + 4\epsilon^2]2\Lambda^2}{\mu\mathcal{R}\mathfrak{K}}\|\nabla F(x_0^s)\|^2
$$

$$
+ \frac{[\eta(1-\alpha_1\beta_1) - 2L\eta^2(1-\alpha_1\beta_1)^2](4\theta\Lambda - \theta^2\Lambda^2 - 2)8\sigma^2}{B\mathcal{R}}. \tag{55}
$$

Further, as defined in SN-GAG (Algorithm 1), $\tilde{x}^s = x_{\mathfrak{K}}^s$ and $\tilde{x}^{s-1} = x_0^s$, we have

$$\mathbb{E}[\|\nabla F(\tilde{x}^s)\|^2] \leq \frac{[4\alpha_2(1-\beta_2^{\mathfrak{K}})\gamma^2 + 4(1-\alpha_2)\gamma^2 + 4\epsilon^2]2\Lambda^2}{\mu\mathcal{R}\mathfrak{K}}\|\nabla F(\tilde{x}^{s-1})\|^2$$
$$+ \frac{[\eta(1-\alpha_1\beta_1) - 2L\eta^2(1-\alpha_1\beta_1)^2](4\theta\Lambda - \theta^2\Lambda^2 - 2)8\sigma^2}{B\mathcal{R}}. \tag{56}$$

In addition, when $\mathfrak{K}$ is slightly large, we have $\beta^{\mathfrak{K}} \to 0$. Therefore, the following result is obtained

$$\mathbb{E}[\|\nabla F(\tilde{x}^s)\|^2] \leq \frac{[4\gamma^2 + 4\epsilon^2]2\Lambda^2}{\mu\mathfrak{K}(4\gamma^2 + 4\epsilon^2)2\eta\Lambda^2 + \mu\mathfrak{K}\mathcal{Q}}\|\nabla F(\tilde{x}^{s-1})\|^2$$
$$+ \frac{[\eta(1-\alpha_1\beta_1) - 2L\eta^2(1-\alpha_1\beta_1)^2](\theta^2\Lambda - 2\theta\Lambda + 2)16\sigma^2}{\eta B(4\gamma^2 + 4\epsilon^2)4\Lambda + B\mathcal{Q}}, \tag{57}$$

where $\mathcal{Q} = [\eta(1-\alpha_1\beta_1) - 2L\eta^2(1-\alpha_1\beta_1)^2](4\theta\Lambda - \theta^2\Lambda^2 - 2)$.

Finally, setting $\eta = \frac{1}{2L(1-\alpha_1\beta_1)}$, $\rho = \frac{2L(1-\alpha_1\beta_1)}{\mu\mathfrak{K}}$, and applying the inequality (57) recursively, we obtain the desired results, i.e., $\mathbb{E}\left[\|\nabla F(\tilde{x}^{\mathfrak{S}})\|^2\right] \leq \rho^{\mathfrak{S}}\|\nabla F(\tilde{x}^0)\|^2$.

## B.2 Proof of Theorem 2

Following the proof of Theorem 1, we can complete the proof of SN-GAG (Algorithm 2) for the PŁ case.

The result in the inequality (54) and Assumption 1(b) result in the following inequality

$$\mathbb{E}[\|\nabla F(x_{\mathfrak{K}}^s)\|^2] \leq \frac{[4\alpha_2(1-\beta_2^{\mathfrak{K}})\gamma^2 + 4(1-\alpha_2)\gamma^2 + 4\epsilon^2]4\Lambda^2\delta}{\mathcal{R}\mathfrak{K}}\|\nabla F(x_0^s)\|^2$$
$$+ \frac{[\eta(1-\alpha_1\beta_1) - 2L\eta^2(1-\alpha_1\beta_1)^2](4\theta\Lambda - \theta^2\Lambda^2 - 2)8\sigma^2}{B\mathcal{R}}. \tag{58}$$

Similarly, considering $\tilde{x}^s = x_{\mathfrak{K}}^s$, and $\tilde{x}^{s-1} = x_0^s$, we further derive

$$\mathbb{E}[\|\nabla F(\tilde{x}^s)\|^2] \leq \frac{[4\alpha_2(1-\beta_2^{\mathfrak{K}})\gamma^2 + 4(1-\alpha_2)\gamma^2 + 4\epsilon^2]4\Lambda^2\delta}{\mathcal{R}\mathfrak{K}}\|\nabla F(\tilde{x}^{s-1})\|^2$$
$$+ \frac{[\eta(1-\alpha_1\beta_1) - 2L\eta^2(1-\alpha_1\beta_1)^2](4\theta\Lambda - \theta^2\Lambda^2 - 2)8\sigma^2}{B\mathcal{R}}. \tag{59}$$

The condition $\beta_2^{\mathfrak{K}} \to 0$ makes the following result be inferred

$$\mathbb{E}[\|\nabla F(\tilde{x}^s)\|^2] \leq \frac{(4\gamma^2 + 4\epsilon^2)4\Lambda^2\delta}{\mathfrak{K}(4\gamma^2 + 4\epsilon^2)2\Lambda^2 + \mathfrak{K}\mathcal{Q}}\|\nabla F(\tilde{x}^{s-1})\|^2$$
$$+ \frac{[\eta(1-\alpha_1\beta_1) - 2L\eta^2(1-\alpha_1\beta_1)^2](4\theta\Lambda - \theta^2\Lambda^2 - 2)8\sigma^2}{2\eta B(4\gamma^2 + 4\epsilon^2)\Lambda^2 + B\mathcal{Q}}. \tag{60}$$

Finally, when adopting $\eta = \frac{1}{2L(1-\alpha_1\beta_1)}$, $\hat{\rho} = \frac{4L\delta(1-\alpha_1\beta_1)}{\mathfrak{K}}$, and applying the inequality (60) recursively, the desired results were obtained.

$$\square$$

