# OpenReview forum: "Adaptive Second-Order Stochastic Optimization"
_ICLR.cc/2025/Conference — ICLR 2025 Conference Withdrawn Submission_

### Official Review · Reviewer_jgam · 2024-10-22

**Soundness:** 2
**Presentation:** 1
**Contribution:** 3
**Rating:** 3
**Confidence:** 3

**Summary:**

This paper proposes two algorithms for second-order stochastic optimization that incorporates ideas from adaptive gradient methods, in particular quasi-hyperbolic momentum. Many popular stochastic optimization algorithms (such as Adam) fall into this framework, and this framework is general enough to be able to utilize various gradient estimators. When the objective functions are strongly convex / satisfy the Polyak-Lojasiewicz property, these algorithms are proven to converge linearly. Numerical experiments are done on logistic regression / SVM objectives confirming the theory and showing that the the proposed algorithms are fairly robust to hyper-parameter settings.

**Strengths:**

- This application of quasi-hyperbolic momentum appears to be novel.
- The proposed algorithms converge linearly, empirically faster than baselines, and are fairly robust to hyper-parameter settings.

**Weaknesses:**

- There is room for improvement in the paper's writing.
  - Although the abstract states that experimental results show the effectiveness of the algorithms, they are left to the appendix without indication, which is confusing.
  - More high-level explanation of how QHM is plugged into adaptive gradient methods to obtain Algorithms 1 and 2 would be helpful.
  - No proof of Theorems 3 and 4 are provided in the appendix.
- Experiments are only done on binary classification problems, whereas many of the discussed optimization methods are used on larger image-scale datasets. It would strengthen the paper to include experiments on some larger problems, like CIFAR100.
- Assumption 2 appears to imply strong convexity. Therefore, it should not appear in Theorem 4.

**Questions:**

- How are the hyperparameters chosen in the experiments?

---

### Official Review · Reviewer_gBuK · 2024-10-23

**Soundness:** 2
**Presentation:** 2
**Contribution:** 3
**Rating:** 5
**Confidence:** 3

**Summary:**

This paper addresses the issue of learning rate setting in the SSO method by developing a class of adaptive SSO algorithms. A convergence analysis of these algorithms is also conducted under two different conditions: SC (Strong Convexity) and PL (Polyak-Lojasiewicz).

**Strengths:**

1. The idea of using adaptive gradients in the SSO algorithm should be novel.
2. The paper provides detailed convergence guarantees under various conditions

**Weaknesses:**

1. There are some formatting issues, such as the incorrect alignment of Assumption 2.
2. It confuses me that in the pseudocode for Algorithms 1 and 2, the value of $k$ starts from 1, but the first-order method only updates the gradient when $k<1$.  It seems to imply that there is no case where the first-order gradient is used alone. This makes it difficult for me to understand the algorithm and proof presented in this paper.
3. I have some doubts about Remark 1(1) because SARAH and SPIDER do not seem to be unbiased algorithms. I believe this point might need further clarification.
4. The proposed algorithm does not seem to show a clear advantage over SVRG in terms of experimental results and theoretical analysis, yet it introduces several additional hyperparameters. I am curious whether it can consistently maintain its advantages under different hyperparameter settings.

**Questions:**

Overall, I believe this paper makes some contribution, but I am taking a cautious stance given the current results. I encourage the authors to address the concerns raised in this review, and if clarified, I would be glad to reconsider my evaluation more favorably.

---

### Official Review · Reviewer_nGwc · 2024-11-01

**Soundness:** 2
**Presentation:** 1
**Contribution:** 1
**Rating:** 3
**Confidence:** 4

**Summary:**

This work proposes and analyzes an adaptive variant of Newton and quasi-Newton steps. Namely, the "base" step is a preconditioned variance-reduced gradient step, where the preconditioning is Newton or quasi-Newton. The authors then propose to use heavy-ball momentum on this base step as well as a vanilla Adam preconditioner (using the squares of the original gradients). Convergence rates and some experiments are provided.

**Strengths:**

- The algorithm does well on LIBSVM datasets.
- The authors describe the algorithms in details, which helps clarity.
- Convergence analysis and experiments provided.

**Weaknesses:**

The writing is not good in general. Sometimes, it is incoherent and verbose, such as the introduction. The text contains unusual word choices and writing style that slightly degrade the credibility of this work. I will mention some examples:
- The opening sentence is a good example. It is very long and poorly written.
- line 038: “is continuously but possibly non-convex”.  Continuous functions are much more likely to be non-convex actually. In addition to that, continuous functions are not necessarily convex (e.g., the indicator function on the extended reals is discontinuous but convex).
- The acronym for "artificial intelligence" is chosen to be AR. Why? It is also used again in line 055.
- Equation 3 is elementary (and thus unnecessary). Also, the tags are wrongly typeset. The authors are suggested to use either (GD) and (SGD) or just (3). The very next sentence “the iterative scheme (3) falls into vanilla SGD” shows the ambiguity of the current tags.
- line 068: “Via automatically acquiring the learning rate for SGD,” seems to use a wrong paraphrasing of "adaptively obtaining/getting". It sounds unnatural to me.
- line 072: “[…] are another continually being discussed and updated technique.” is grammatically wrong and incoherent.
- line 072-073: “From the side of manipulating variance of […]”. The word “manipulating” in this context sounds weird, e.g., “”reducing” sounds more natural.
- line 079 “solve the impractical of evaluating” is grammatically wrong.
- The citation for the Adam paper is missing Jimmy Ba. It is concerning that the authors did not notice this, given that they have cited this paper multiple times.
- Assumption 2 looks as if it was included in Assumption 1 because the indentation is wrong.
- The \mathfrak style letters used for the total number of inner/outer iterations look out of place. Why not just use S and K?
- Algortihm 1 and 2 are almost equivalent. The only difference is that the first uses Newton whereas the second uses Quasi-Newton. This is also mentioned by the authors in the text. Thus, Algorithm 2 is redundant, especially since it’s taking half a page, so why waste this precious space? Or rather, why not use this chance to make the paper more concise and get it closer to the recommended 9-page length?

Based on the quality of the writing alone, I highly recommend the authors to reconsider this submission and put some effort into polishing it. Nonetheless, I will further support my decision to reject this paper based on its content as well.

- The method uses a preconditioned variance-reduced gradient step, with (stochastic) Newton or quasi-Newton preconditioning. Heavy-ball momentum is then applied on this step and then vanilla Adam preconditioner. The double use of preconditioners and momentum (the latter of which amounts to a rescaled learning rate and momentum parameters) is not exactly indicative of a deep understanding of adaptive gradient methods, which is a bit worrisome.
- Assumption 2 implies that *each* function is strongly convex and smooth, making this assumption a bit too restrictive, i.e., $f(x,\xi)$ is strongly convex and smooth for *all* $\xi$.
- Experiments are done only on LIBSVM, which is not sufficient to demonstrate the effectiveness of the proposed method. More importantly, there are no Newton/quasi-Newton baselines. The comparison with adaptive gradient methods is not fair in terms of computational budgets. The authors should make the running time clear in the figures (specifically, Figure 1), showing the plots in terms of wall time or effective passes, i.e., number of times a forward/backward pass is done, where second-order (Newton) methods use quadratically as many passes as first-order (gradient) methods.
- Supplying the (anonymous) code for this work could have helped the reviewers understand the implementation in minute details for further assessment, but no supplementary materials nor any anonymous repository were provided.

**Questions:**

None.

---

### Official Review · Reviewer_nuZX · 2024-11-02

**Soundness:** 2
**Presentation:** 2
**Contribution:** 1
**Rating:** 3
**Confidence:** 5

**Summary:**

This paper proposes a class of adaptive second-order stochastic optimization methods, which include one General Adaptive Gradient Stochastic Newton (GAG-SN) method and one General Adaptive Gradient Stochastic Quasi-Newton (GAG-SQN) method. Moreover, the authors also provided the convergence behavior and computational complexity for the proposed methods. Some experimental results are reported.

**Strengths:**

The paper is complete in format.

**Weaknesses:**

1.	This paper mainly considers the problem (2). But there are many recently proposed first-order convex optimization algorithms such as Katyusha [1] and non-convex optimization methods such as Spider [2] and SpiderBoost [3] for convex and non-convex problems (2).
[1] Zeyuan Allen-Zhu. Katyusha: The First Direct Acceleration of Stochastic Gradient Methods.
[2] Cong Fang, Chris Junchi Li, Zhouchen Lin, Tong Zhang. SPIDER: Near-Optimal Non-Convex Optimization via Stochastic Path-Integrated Differential Estimator.
[3] Zhe Wang, Kaiyi Ji, Yi Zhou, Yingbin Liang, Vahid Tarokh. SpiderBoost and Momentum: Faster Stochastic Variance Reduction Algorithms.
What’s the advantage of the proposed second-order methos against existing first-order methods?
2.	There is no comparison of the convergence rates of the proposed algorithms and existing first-order methods for the cases of strongly-convex, general convex and non-convex.

**Questions:**

1.	The experimental results are not convincing. The authors should compare the proposed algorithms with more recently proposed algorithms in terms of running time.
2.	The dimensions of all the used data sets in this paper are very smaller. Therefore, the authors should report the results of high-dimensional problems on more data sets such as rcv1.

---

### Official Review · Reviewer_eTMt · 2024-11-03

**Soundness:** 2
**Presentation:** 3
**Contribution:** 2
**Rating:** 5
**Confidence:** 4

**Summary:**

This paper tackles the challenge of learning rate adaptation in second-order stochastic optimization (SSO) methods, an area that has received limited research attention despite its importance. The authors introduce a general adaptive gradient (GAG) framework with quasi-hyperbolic momentum, which encompasses popular methods like Adam and AdaGrad, and combine it with stochastic Newton and quasi-Newton methods to create SN-GAG and SQN-GAG. They provide a unified theoretical analysis covering convergence under both strongly convex and Polyak-Łojasiewicz conditions, with the latter being a novel contribution to the field. Experimental results demonstrate the superior performance and robustness of these new methods in machine learning applications.

**Strengths:**

1. The paper presents its ideas in a clear, structured manner with logical flow between sections.
2. The authors provide rigorous theoretical analysis proving convergence under both strongly convex functions and the more general Polyak-Łojasiewicz condition.

**Weaknesses:**

1. A significant limitation of the empirical evaluation is its narrow scope, focusing only on binary classification tasks and strongly-convex scenarios, which may not fully demonstrate the methods' effectiveness across diverse machine learning problems.
2. The theoretical analysis relies on numerous restrictive assumptions regarding gradient and Hessian norm bounds (Assumptions 1 & 2), potentially limiting the practical applicability and generality of the proposed theorems.
3. The paper would benefit from a more comprehensive comparison with recent advances in second-order stochastic optimization, such as AdaFisher and AdaHessian. A broader literature review and comparative analysis would better position this work within the current state-of-the-art.

[1] AdaFisher: Adaptive Second Order Optimization via Fisher Information
[2] Adahessian: An adaptive second order optimizer for machine learning

**Questions:**

I have no questions.

---

### Note · Authors · 2024-11-22

I have read and agree with the venue's withdrawal policy on behalf of myself and my co-authors.